# LEAP-Based Greenhouse Gases Emissions Peak and Low Carbon Pathways in China’s Tourist Industry

**DOI:** 10.3390/ijerph18031218

**Published:** 2021-01-29

**Authors:** Dandan Liu, Dewei Yang, Anmin Huang

**Affiliations:** 1College of Tourism, Huaqiao University, Quanzhou 362021, China; 00_liu@sina.cn; 2School of Geographical Sciences, Southwest University, Chongqing 400715, China; younglansing@gmail.com

**Keywords:** GHG peak, LEAP model, low carbon pathways, scenarios, tourist industry

## Abstract

China has grown into the world’s largest tourist source market and its huge tourism activities and resulting greenhouse gas (GHG) emissions are particularly becoming a concern in the context of global climate warming. To depict the trajectory of carbon emissions, a long-range energy alternatives planning system (LEAP)-Tourist model, consisting of two scenarios and four sub-scenarios, was established for observing and predicting tourism greenhouse gas peaks in China from 2017 to 2040. The results indicate that GHG emissions will peak at 1048.01 million-ton CO_2_ equivalent (Mt CO_2e_) in 2033 under the integrated (INT) scenario. Compared with the business as usual (BAU) scenario, INT will save energy by 24.21% in 2040 and reduce energy intensity from 0.4979 tons of CO_2_ equivalent/10^4^ yuan (TCO_2e_/10^4^ yuan) to 0.3761 Tce/10^4^ yuan. Although the INT scenario has achieved promising effects of energy saving and carbon reduction, the peak year 2033 in the tourist industry is still later than China’s expected peak year of 2030. This is due to the growth potential and moderate carbon control measures in the tourist industry. Thus, in order to keep the tourist industry in synchronization with China’s peak goals, more stringent measures are needed, e.g., the promotion of clean fuel shuttle buses, the encouragement of low carbon tours, the cancelation of disposable toiletries and the recycling of garbage resources. The results of this simulation study will help set GHG emission peak targets in the tourist industry and formulate a low carbon roadmap to guide carbon reduction actions in the field of GHG emissions with greater certainty.

## 1. Introduction

Climate change has unprecedented impacts on the world such as food reduction, glacier melting and the rise of sea levels [1]. Tourism is closely related to climate change and is one of the key drivers of energy consumption and greenhouse gas (GHG) emissions [2]. For many years, the complex relationship between tourism and GHG emissions has become a hotspot of tourism research. As the country with the largest GHG emissions, China plays a crucial role in resisting climate change [3] and has promised to reach its peak total of GHG emissions by 2030 [4]. Tourism significantly contributes to the national economy [5]. The quantity of energy consumption resulting from its development has continued to rise and the pressure on emission reduction has also grown immensely [6]. The time peak and the pace of the reduction of tourism emissions are critically important to the Chinese goal [7]. Reducing tourism GHG emissions not only helps to offset global warming but also contributes to sustainable tourism development [8]. A consensus has been reached to curb tourism GHG emissions. However, a general lack of understanding exists regarding the relationship between the peak of tourism GHG emissions and the policy-driven emission reduction. This study is an attempt to bridge this gap.

Tourism is not a traditional sector in the System of National Accounts [8] and the lack of tourism statistics makes it difficult to calculate carbon emissions from tourism [9]. However, tourism-related carbon emissions remain a central issue of several studies [10]. The measurement of tourism carbon emissions can generally be divided into two categories. The first category of research mainly focuses on the calculation of carbon emissions from tourism using methods such as the bottom-up approach [11], life cycle assessment [12] and the carbon footprint approach [9]. The study objects are mostly single or multiple tourism elements including tourism transportation, tourism accommodation and tourism attractions. Gössling [13] was the first to measure tourism-related carbon emissions and found that they relied heavily on energy consumption especially for air transport. Becken and Patterson [14,15,16,17] asserted that different travel choices within the accommodation, tourism activity and transportation subsectors demand different amounts of energy. Wu and Shi [18] calculated the carbon emissions of transportation, catering, accommodation and tourism activities in China. The second category of research primarily emphasizes the calculation of carbon emissions from tourism using methods such as the top-down approach [10], the input–output method and the Tourism Satellite Account (TSA). Meng et al. [19,20] used the TSA and the input–output model to measure direct and indirect carbon emissions in China. Tang and Ge [10] employed Shanghai as an example for calculating the carbon emissions that result from tourism consumption using the input–output model. The results demonstrate that tourism is not a low carbon industry. The prediction of the total energy consumption and the GHG emission intensity is the key point of energy development planning and environmental protection. In terms of tourism carbon emissions, many scholars have made future predictions according to the analysis of current emissions and the development of the tourism economy market. Dubois and Ceron [21] predicted the French tourism mobility demand in 2050 and associated GHG emissions. They found that French tourism GHG emissions could increase by 90% in 2050 without emission reduction measures. Peeters and Dubois [22] presented a 30-year projection and a 45-year simulation in tourism and confirmed that improvements in technology alone are insufficient as major shifts in transport modes and destination choice (less distant) are necessary. Although the nexus between carbon emissions and tourism has been widely studied over time, the measurement of tourism carbon emissions remains controversial. First, the bottom-up analysis requires detailed information on energy use and tourism behaviors. There is a lack of information on carbon emissions because only a few tourism sectors are involved. Second, although the top-down method comprehensively measures the carbon emissions of the tourist industry, it has a shortcoming pertaining to the lack of an annual input–output table of China such that the yearly carbon emissions of the tourist industry cannot be calculated. Moreover, using the direct consumption coefficient of the input–output method for calculating indirect tourism carbon emissions is inappropriate. Finally, the peak prediction and scenario study of tourism carbon emissions is an important premise for measuring the impact of tourism on global environmental change and is also a scientific basis for tourism to slow down and respond to global environmental change. However, few studies involve the peak prediction of tourism carbon emissions.

This work has taken Chinese tourism as its research base. The life cycle assessment method and material flow theory were adopted, coupled with the calculation and analysis of the total energy consumption and GHG emissions and the use of the long-range energy alternatives planning system (LEAP) model to forecast GHG emissions from the energy consumption of the tourist industry from 2017 to 2040. The LEAP model was developed at the Stockholm Environment Institute for energy policy analysis and climate change mitigation assessments. The LEAP is a bottom-up forecasting model that allows integrated resources planning, GHG mitigation assessments and the development of low emission strategies [23,24]. The model can be employed for the long-term prediction and scenario analysis of energy demand and the associated environmental problems. Given its advantages in alternative predictions, accuracy and policy settings, the LEAP has been widely applied in energy strategy study at a national, regional and sectoral scale [25,26,27,28]. 

Therefore, in this study, we address these challenges in tourism carbon emissions research by making two important contributions. First, this work develops a novel approach to assess the peak time and pace of reduction of tourism carbon emissions according to a credible national policy [7]. The peak prediction for tourism carbon emissions in China and the optimization of the energy conservation and emission reduction path in the tourist industry with the LEAP model is powerful in the scenario analysis of policy quantification and at a technical level. Second, this work explores new perspectives and develops a better understanding of tourism GHG emissions in its energy flow at different stages of the life cycles in the hopes of bridging the gaps found in previous research on GHG emissions of tourism. The LEAP model is utilized to calculate and simulate tourism GHG emissions including the life cycle process and the mechanism of energy from mining, transportation, processing, conversion, transmission, distribution, terminal utilization to waste as well as the redefinition of indirect tourism carbon emissions. This article also supplements and enriches the application of the LEAP model at an industrial level especially in a non-energy leading industry such as tourism and provides reference for other industries with similar attributes (such as the cultural industry) in studying energy consumption and carbon emissions.

## 2. Materials and Methods

### 2.1. LEAP-Tourist Model

The LEAP-Tourist model developed in this paper is in line with the specific situation of tourism in China. The model takes 2017 as the baseline year and 2040 as the final year. The analysis consists of three modules: the end-use demand, energy transformation and energy supply. According to the Energy Balance Table and other relevant statistical data, the end-use demand module consists of transportation, accommodation, catering, sightseeing, shopping and waste material sectors. The energy transformation sector of tourism is divided into power generation, coal washing, coking, gas making, coal products processing, oil refining, heat supply and so on. The energy supply module includes primary and secondary energy. In the LEAP-Tourist model, the primary energy resources mainly consist of bituminous coal, biomass, solar, hydro, wind and natural gas. The secondary energy resources are gasoline, coke, liquefied petroleum gas, diesel, electricity and heat.

The results of the LEAP model include predictions of the energy demand and environmental impact. The research flowchart based on the LEAP-Tourist model is shown in Figure 1.

### 2.2. Scenario Design

To analyze the environmental effects resulting from energy saving and the transformation policies implemented in tourism, two scenarios were considered in this study: the business as usual (BAU) scenario and the integrated (INT) scenario. The BAU scenario refers to the continuation of current energy saving and emission reduction policies and measures on the premise of achieving the established socio-economic developmental goals. The BAU scenario is the baseline reference for the INT scenario. The INT scenario is divided into four sub-scenarios: the department structure optimization (DSO) sub-scenario, the clean energy substitution (CES) sub-scenario, the energy saving facilities (ESF) sub-scenario and the low carbon behavior (LCB) sub-scenario. The recommended measures for the two scenarios and four sub-scenarios are listed in Table 1.

### 2.3. Time Series Trend Prediction Method

The GHG calculation method of the LEAP model is in accordance with the data of energy consumption in different sectors. Therefore, investigating and computing the historical data of tourism sub-sectors and selecting the appropriate prediction model to predict the economic development trend of tourism before 2040 are necessary. The combination forecasting model of a time series seeks to combine the results of different prediction models with a certain weight and cannot reflect the interaction relationship of each subsystem in the source and process. This study optimizes the existing time series trend model through theoretical assumptions and mathematical derivation and performs prediction analysis on the industrial structure of the tourist industry.

### 2.4. Data Collection

The data of the end-use demand module of the LEAP-Tourist model serve to: (1) identify the income of various tourism sectors in 2017 through the China Statistical Yearbook 2018, (2) establish the corresponding relationship between tourism sectors and national economic industries and calculate the energy consumption of tourism sectors through the ratio of the added value of each sector and corresponding industries and (3) acquire details from the tourism waste sector through the ratio of the total tourism revenue to the gross domestic product (GDP).

In addition to the direct consumption of the end-use consumer sector, the products produced by the processing conversion module also involve the relationship of processing and re-input. The calculation idea is as follows: (1) allocate the quantity of each energy from local production, import, export, inventory and conversion traced back with the knowledge of each energy demand of the tourism consumption sector to the tourist industry in proportion to obtain the quantity of each energy from the processing conversion of the tourist industry; (2) determine the conversion process that produces energy, the needed energy and the conversion efficiency and (3) calculate the resource supply and energy data required for the processing and conversion of the tourism consumption sector.

The policy basis for scenario setting comes from the China Low Carbon Yearbook 2017, China Low Carbon Cycle Yearbook 2018, guidance on further promoting energy conservation and emission reductions in the tourist industry, several opinions on the tourist industry’s response to climate change, 100 items of hotel energy conservation and emission reductions and 30 items of energy conservation and emission reductions of level scenic spots. All environmental emission factors in this study are from the Intergovernmental Panel on Climate Change(IPCC) 1 default emission factors embedded in the LEAP model. The main GHGs include methane (CH_4_), nitric oxide (N_2_O) and carbon dioxide (CO_2_).

## 3. Tourism Revenue Forecast

### 3.1. Order Parameter Identification

In this study, the order parameter was used to measure the state of the tourist industry, an aspect that has an important impact on the structure and operation trend of the tourist industry and is reflected in the comprehensive effect of all elements on the collaborative application of the system. Therefore, the tourism revenue and sector structure were taken as the order parameters of this work. The research years of the selected order parameters were 2011–2017. As shown in Table 2.

### 3.2. U Value Calculation

In this paper, *u* is characterized as the degree of the self-organization of the system and is associated with the time series model so as to give some meaning to the intermediate variable *u* in the n-dimensional order parameter equation.
Order *u* = *f*(*t*), *u*∈(0,1), when *t*→0, *u*→0; when *t*→+∞, *u*→1.(1)

Let *u* and *t* satisfy the following relation:*u* = *f*(*t*) = 1 − *e*^−*at*^, *u* ∈ (0,1)(2)
where *a* is a parameter and *a* > 0, *a* can be set to 0.05 by default. It can be set according to the research time span of the trend model. When the time span is short, *a* takes the larger value. When the time span is large, *a* takes the smaller value. According to Formula (1), the calculation results of *u* are shown in Table 3.

### 3.3. Trend Forecast

Although linear regression can meet the vast majority of data analysis, it cannot be applied to all data in reality. The relationship between dependent variables and independent variables cannot determine whether it is a linear or other non-linear model. It needs to use curvilinear regression to determine the most suitable model between variables. Curvilinear regression refers to the method of regression analysis for non-linear variables. We analyzed the data from 2011 to 2017 and used SPSS software to estimate the curves of *u* and tourism revenue. The types of curve estimation included linear, logarithmic, reciprocal, quadratic, cubic, power, S and exponential curves. The parameter estimates are shown in Table 4.

Analysis of the results of the above curve fitting indicated that the *R*^2^ of the cubic model was the largest, thereby suggesting the highest degree of fit under the existing data. The significance was also 0.00. It showed that for the relationship function expression between the independent variable *u* and the dependent variable tourism total income in this study, the most suitable was the cubic function expression:
*Y* = 21678.82 − 1861.93 * *u* + 448942.86 * *u*^2^ − 237473.50 * *u*^3^.(3)

The value of *u* from 2018–2040 by Formula (1) was calculated and substituted into Formula (2) to obtain the tourism income of the study year. The China Statistical Yearbook indicated that the total revenue of the tourist industry in 2018 and 2019 were CNY 5970 billion and CNY 6630 billion, respectively. The calculated forecast values were CNY 6135.08 billion and CNY 6865.63 billion, with comparison errors at 2.76% and 3.55%, respectively. Thus, the results of the prediction model were relatively accurate and could be used to predict the macro order parameter value of the tourist industry.

## 4. Results

### 4.1. Energy Demand Forecast

The total energy consumption forecast in tourism from 2017 to 2040 under the BAU and INT scenarios are shown in Figure 2. Under the BAU scenario, tourism’s energy consumption will increase by 2.32 times from 268.87 million-ton coal equivalent (Mtce) in 2017 to 892.36 Mtce in 2040. In 2017, the energy intensity was 0.4979 ton coal equivalent/10^4^ yuan (Tce/10^4^ yuan) and decreased to 0.4962 Tce/10^4^ yuan in 2040. After implementing a series of energy saving and emission reduction policies and measures, the energy consumption under the INT scenario decreased significantly. In 2040, tourism’s energy consumption would be 676.35 Mtce, indicating a reduction of 216.01 Mtce and 24.21% of energy savings compared with the BAU scenario. At the same time, the energy intensity decreased to 0.3761 Tce/10^4^ yuan. 

Under the two scenarios and four sub-scenarios, the proportion of fuel types in tourism’s energy system would drastically change by 2040 with the clean energy substitution sub-scenario being the most obvious change (see Figure 3.). From the perspective of fuel consumption, gasoline and diesel accounted for 19.67% and 37.52% of the total energy consumption in 2017 under the BAU scenario, respectively. By contrast, the proportion of gasoline and diesel would decrease to 14.85% and 28.31% of the total energy consumption in 2040 under the INT scenario, respectively. Moreover, the proportion of electricity, wind and solar energy would rise significantly. The increase of clean energy sources could contribute to the improvement of the regional environment.

From the perspective of sector consumption, the energy demand in the BAU scenario in 2017 mainly came from the transportation sector, as shown in Figure 4. In the INT scenario, the transportation sector still accounted for the largest proportion but would drop from 88.47% in 2017 to 82.82% in 2040. This outcome was due to the effect of the optimization scenario of the departmental structure and depends on the development of China’s railways. The proportion of passenger railway transportation is also increasing. The gradual increase in the proportion of energy consumption in the accommodation, catering and shopping sectors indicated that the expenditure in the accommodation, catering and shopping sectors increased as a proportion of the total tourism expenditure. Thus, the high-quality development of tourism would meet the growing needs of people for a better life.

### 4.2. GHG Emissions

Figure 5 shows the prediction of GHG emissions in tourism from 2017 to 2040 under the BAU and INT scenarios. Under the BAU scenario, the GHG emissions of tourism in 2040 would be 1798.67 million-ton CO_2_ equivalent (Mt CO_2e_), a figure that is two times higher than that in 2017. Under the INT scenario, the GHG emissions in tourism would decrease significantly after the initial increase. The peak in emissions is predicted to occur in 2033, at 1048.01 Mt CO_2e_. In 2040, GHG emissions would drop to 1012.87 Mt CO_2e_. The intensity of the GHG emissions would continue to decline from 1.1098 tons of CO_2_ equivalent/10^4^ yuan (TCO_2e_/10^4^ yuan) in 2017 to 1.0002 TCO_2e_/10^4^ yuan in 2040 under the BAU scenario and would continue to decline under the INT scenario to only 0.5632 TCO_2e_/10^4^ yuan in 2040, a decrease of 43.87% compared with the BAU scenario.

In terms of the proportion of GHG emissions by sector, the transportation sector contributed the most in 2040 under the BAU scenario and accounted for 84.70%, as shown in Figure 6. Under the INT scenario, the proportion contributed by the transportation sector would decrease to 80.21%. In addition to the decline in the proportion of GHG emissions in the transportation sector, the proportion of other sectors increased. Among them, the GHG emissions from the waste sector rose the most. The GHG emissions of tourism waste cannot be ignored but this aspect has been rarely included in previous studies.

### 4.3. Emission Reduction Contribution

Table 5 lists the contribution of GHG emission reductions in each sub-scenario and sector. If all policies and measures were implemented, then the potential of emission reduction in tourism would gradually change from 65.71 Mt CO_2e_ in 2020 to 785.79 Mt CO_2e_ in 2040. Regarding the emission reduction contribution of each sub-scenario, the CES sub-scenario would make the largest contribution followed by the LCB sub-scenario. From the sectoral perspective of emission reductions, the transportation sector shows the largest contribution followed by the catering sector but the proportion would decrease annually. The proportion of waste emission reductions was small but increased annually.

## 5. Discussion

As demonstrated by the numerous research on the modeling and projection of tourism GHG emissions, a better understanding of future tourism GHG emissions is exceedingly important for addressing the sustainable development of tourism and the feasibility of the GHG reduction goal. This study established a LEAP-Tourist model to project and analyze the energy structure, energy consumption and GHG emissions under different policies in tourism and provides a helpful supplement to existing research methods and frameworks. In the Anthropocene, far-reaching national policies by the largest players may become triggers of climate change. Previous studies were mostly based on econometrics, an approach that cannot reflect the changes generated by energy saving measures in the tourist industry. These predicted results are always higher than the actual results. Therefore, only by continuously strengthening the understanding of the relationship between tourism economy, energy saving measures and technology can we grasp the demand trend more accurately so as to identify the optimal path to guide the current development direction and mode of the tourist industry in China.

CO_2_ is the main GHG emission gas and also the main culprit of global warming. Most studies employ CO_2_ as the only research object [19,29,30,31]. In this study, the proportion of CO_2_ exceeded 96% in the BAU scenario but decreased in the INT scenario. However, the role of other GHGs should not be underestimated. Given the limitation of the carbon source and the simplicity of calculation, the current research on tourism carbon emissions only calculated the CO_2_ emissions and research on other types of GHG is insufficient. The countermeasures taken by different GHGs vary. Only by improving the research of the entire GHG can we put forward feasible countermeasures for tourism carbon emissions. CH_4_ is mainly caused by the tourism waste sector. CH_4_ is a more active GHG than CO_2_ and its impact on climate warming is many times higher than that of CO_2_. Under the INT scenario, the proportion of methane gradually increased. Therefore, saving resources and reducing the production of tourism waste are effective means of suppressing CH_4_. The content of N_2_O was also lowest because the important emission source of N_2_O is water, which comes from the extensive use of farmland chemical fertilizer and sewage discharge. Thus, N_2_O in tourism primarily arises from tourism activities. To reduce the production of tourism waste and strengthen rural tourism, the environmental awareness of leisure agriculture is particularly important.

Total GHG emissions in the INT scenario will peak in 2033 and no obvious peak might occur in the BAU scenario. In previous studies, population and economic growth are recognized as important factors in driving CO_2_ emission growth [3,32]. This study also confirmed that tourism economic growth is a major factor in GHG emission growth [5]. Maximal GHG emission reduction also occurred because of the replacement of clean energy in the conversion sector and the tourism sector [33]. The role of clean energy is another crucial factor in determining the peak period to help formulate a feasible timetable action plan on the basis of national goals. The peak time set by China is 2030 whereas the peak time of tourism in this study is 2033. Tourism in China might reach the GHG peak later than the national goal given the continued rapid growth of the tourist industry and soft emission reduction measures. In 2021, China will enter the 14 Five-Year Plan, for which stricter emission reduction measures will be adopted. 

At the beginning of 2020, COVID-19 had a great impact on the global tourist industry [34]. The lockdown measures by China to cope with COVID-19 pushed many small and medium-sized tourism industries on the verge of bankruptcy. Therefore, according to the actual situation, the GHG emissions of the tourist industry in 2020 will be lower than the predicted value in this study. But, at this time, tourism demand was only temporarily suppressed and had not disappeared. With the control of the epidemic situation, the tourism industry in China is recovering gradually and the National Day holiday in October received 637 million domestic visitors [35]. The GHG emissions of the tourist industry will still show an increasing trend in the future. Of course, the new demand and momentum brought by the epidemic will also affect the tourist industry. The innovation of science and technology represented by the internet, big data and artificial intelligence forces the reform of the tourist industry. The “cloud tourism” promoted by live broadcasting has greatly enriched people’s daily life and leisure activities. The series of reforms brought about by this epidemic are undoubtedly green. 

In addition to the CES sub-scenarios, the LCB and DSO sub-scenarios also play significant roles in energy conservation and emission reduction in the tourist industry [6]. The transportation sector with the largest emissions is still the focus of emission reduction [29,36,37]. Thus, the following suggestions are provided. (1) The transportation sector should actively promote the application of various clean fuel vehicles and encourage the development of energy saving and environmentally-friendly vehicles. (2) Traffic control can be implemented in scenic areas, tourists must be encouraged to walk and cycle and transfers between scenic areas should be arranged with battery-powered cars [34,38]. (3) The selection of low carbon tourism methods by tourists should be encouraged. Tourists should bring their own garbage bags as much as possible to resist over-packaged goods and reduce the amount of waste garbage and resource waste. (4) Scenic areas can enhance tourists’ environmental awareness and sense of responsibility for energy conservation and emission reduction through demonstration and guidance, the display of slogans and the distribution of publicity materials [8].

## 6. Conclusions 

Using the LEAP model, a LEAP-Tourist model on energy consumption and GHG emissions for the tourist industry was developed in this study. The model allowed for the accurate calculation and prediction under the influence of different policy measures including energy consumption structure, GHG emissions and the time and scale of the peak and structure of tourism sectors in the energy flow at different phases of the life cycle. These results can help enrich the theories of sustainable development of tourism and low carbon tourism. The outcomes will facilitate the setting of tourism GHG emission peak targets and the development of a low carbon roadmap to address the broader field of non-GHG emissions.

In terms of energy consumption, the energy consumption of the tourist industry under the BAU scenario increased by 2.32 times and mainly involved gasoline and diesel oil. The tourism and transportation sector have the largest energy consumption. Under the INT scenario, the energy saving was 24.21%. 

In terms of GHG emissions, the GHG emissions of the tourist industry under the BAU scenario would double in 2040. By contrast, the GHG emissions of the tourist industry under the INT scenario decreased significantly and showed a trend of an initial increase and then a decrease, with a peak in 2033. 

In terms of energy conservation and emission reduction measures, the contribution rate of the CES sub-scenario and transportation sector was the largest. The transportation sector should pay attention to the application of clean energy, an approach that is an important measure to control air pollution, and explore low carbon tourism. Scenic spots and tourists must strengthen environmental protection awareness and reduce resource waste.

In short, the concept of low carbon tourism development has now been accepted. GHG prediction can better set the peak target of tourism and the pace of emission reduction and provide more effective suggestions for the sustainable development of tourism. Just as any case study, there are a few limitations and possible improvements for future study. (1) Data are still the main restricting factors of tourism carbon emission research. Due to the difficulty of data acquisition, this study used tourism income as the main driving factor. In the next step of research, it can obtain the specific technical parameters of different types of tourism, catering, accommodation, transportation such as the number of beds, electrical type, power consumption and so on. (2) A carbon sink will be included in the measurement of tourism carbon emissions such as water wetland, woodland, arable land, pastoral, grassland and other natural resources as well as the carbon compensation of tourism stakeholders. (3) It is necessary to consider the cost and technical difficulty of emission reduction measures and to improve the carbon neutral path of the tourist industry from economic, social and environmental aspects.

## Figures and Tables

**Figure 1 ijerph-18-01218-f001:**
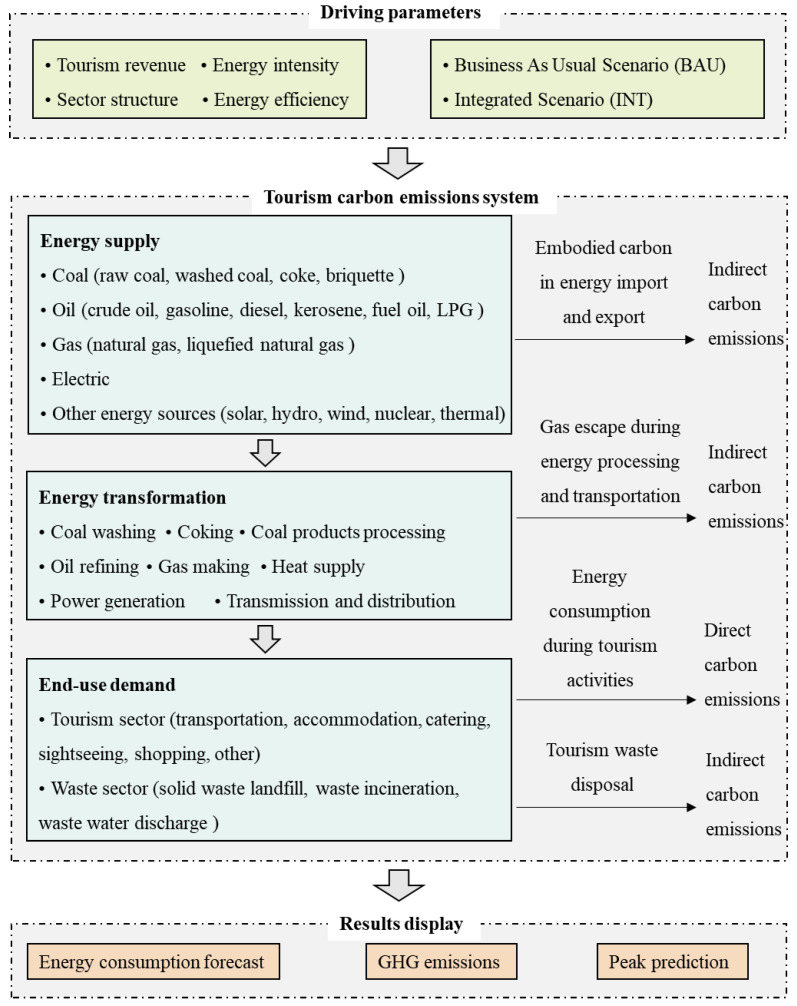
Research flowchart according to the long-range energy alternatives planning system (LEAP)-Tourist model.

**Figure 2 ijerph-18-01218-f002:**
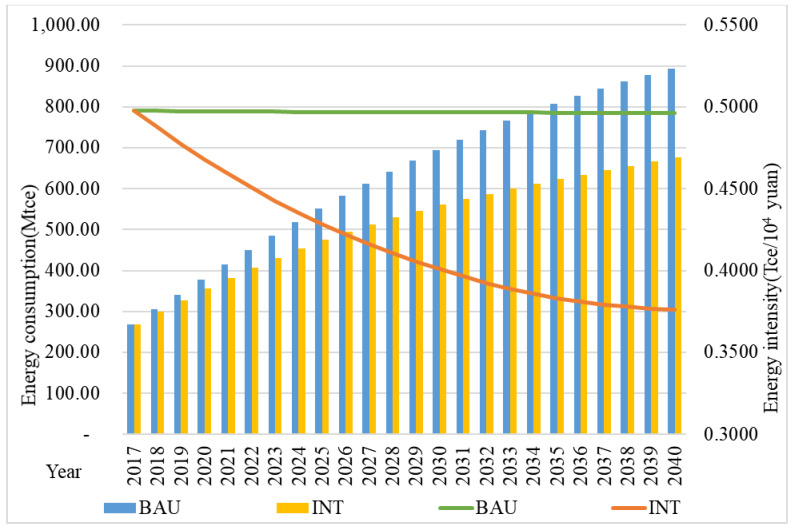
Forecast of total energy consumption and energy intensity under two scenarios.

**Figure 3 ijerph-18-01218-f003:**
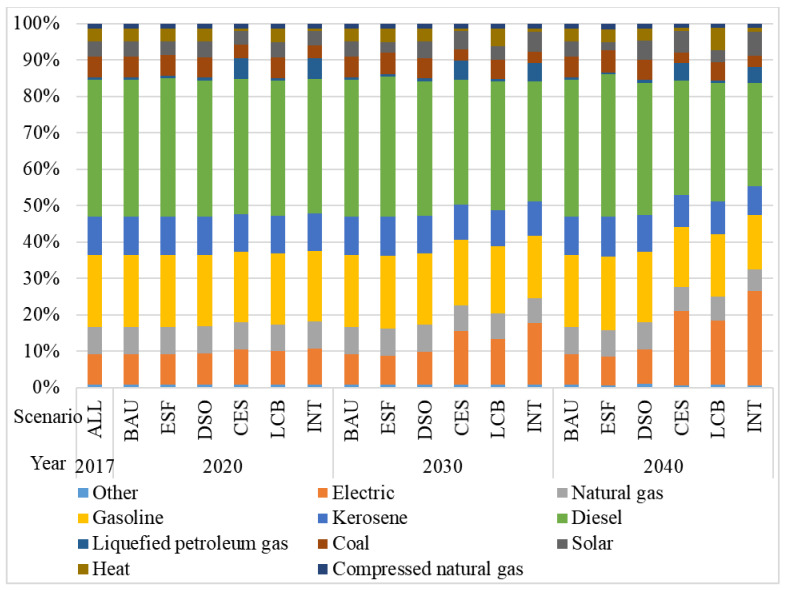
Change of fuel consumption demand under various scenarios.

**Figure 4 ijerph-18-01218-f004:**
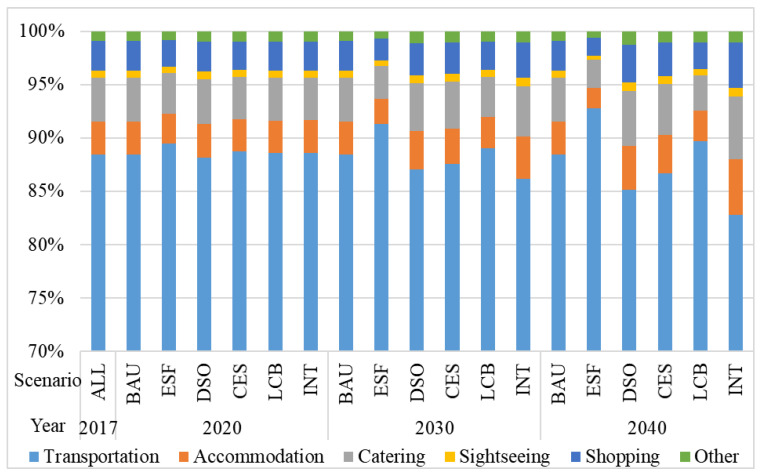
Percentage of total consumption in the energy consumption sector under the various scenarios.

**Figure 5 ijerph-18-01218-f005:**
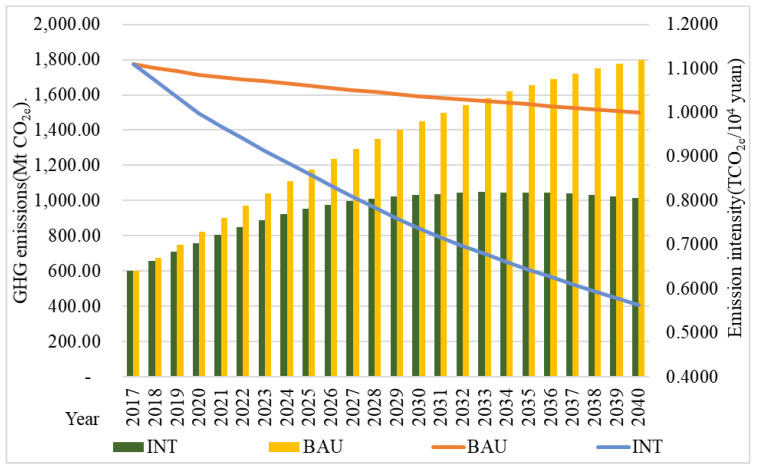
Prediction of greenhouse gas (GHG) emissions and emission intensity under the two scenarios.

**Figure 6 ijerph-18-01218-f006:**
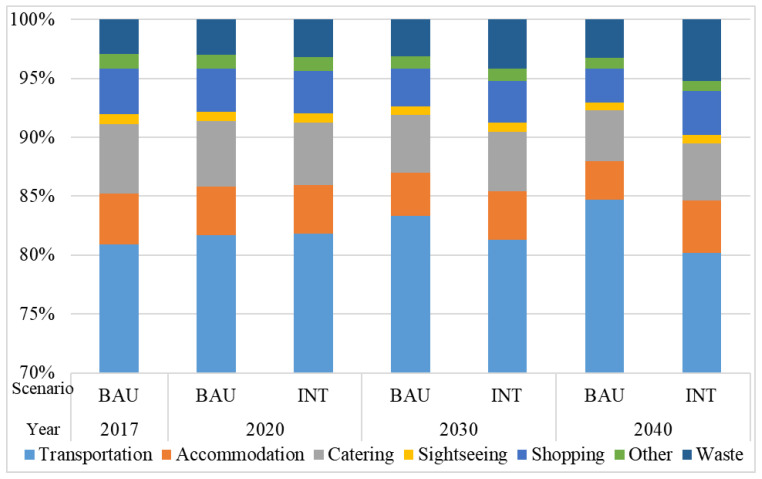
Proportion of GHG emissions by sector under the two scenarios.

**Table 1 ijerph-18-01218-t001:** Policy options and assumptions for four sub-scenarios.

	Scenarios	Department Structure Optimization ^a^				Clean Energy Substitution ^b^				Energy Saving Facilities ^c^	Low Carbon Behavior ^d^
Sectors		2017	2020	2030	2040	2017	2020	2030	2040		
Transportation		36.60%	36.00%	34.00%	30.00%	8.17%	10.00%	15.00%	20.00%	-	1.0% annual reduction of fossil energy intensity, increase clean energy intensity by 2% annually
Accommodation		15.60%	16.00%	17.00%	18.00%	52.68%	60.00%	70.00%	75.00%	2.0% annual reduction of energy intensity	1.0% annual reduction of energy intensity
Catering		20.99%	21.00%	21.50%	22.00%	52.76%					2.0% annual reduction of energy intensity
Shopping		13.93%	14.00%	14.50%	15.00%	52.70%					1.0% annual reduction of energy intensity
Other		7.59%	8.00%	8.50%	9.00%	-					-
Sightseeing		5.29%	5.50%	5.80%	6.00%	-					1.0% annual reduction of energy intensity
Waste		-				-				Reduce waste by 20% compared with the BAU scenario	Reduce waste by 10% compared with the BAU scenario
Power Generation		-				73.50%	60.00%	35.00%	20.00%	-	-
Heat Supply		-				100.00%	80.00%	60.00%	40.00%	-	-

Notes: ^a^ The department structure optimization sub-scenario means the adjustment of the industrial structure and an additional increase in the consumption proportion of catering, accommodation, shopping and sightseeing. ^b^ The clean energy substitution sub-scenario refers to the increase of clean energy consumption in total energy resources. Examples include vigorously promoting new energy vehicles along with promoting solar, wind, geothermal, natural gas and other clean energy heating and lighting applications as well as the shift of power generation and heat supply from coal to wind, solar and other renewable energy sources. ^c^ The energy saving facilities sub-scenario means the energy saving transformation of heating, cooling, ventilation, lighting, refrigeration and other systems; the increase in the proportion of electronic tickets; a reduction of paper tickets; the limit of the use of disposable items and the reduction of resource waste. ^d^ The low carbon behavior sub-scenario promotes green tourism and chooses low carbon travel methods such as walking, cycling and public transportation and reduces the use of disposable daily necessities.

**Table 2 ijerph-18-01218-t002:** Statistics of the tourism order parameters from 2011–2017(Unit: CNY 100 million).

Year	Transportation	Accommodation	Catering	Sightseeing	Shopping	Other	Total tourism revenue
2017	19,766.35	8424.49	11,336.77	2857.42	7521.11	4097.55	54,003.69
2016	16,550.49	7408.16	10,965.71	2869.72	6589.83	2973.20	47,357.10
2015	13,407.81	5927.92	9458.00	2141.36	8127.07	2213.96	41,276.12
2014	10,982.70	4826.71	8298.93	1975.57	6311.56	1404.85	33,800.33
2013	9574.31	3688.86	7224.03	1545.25	6033.56	1413.20	29,479.21
2012	8314.59	3384.57	5960.67	1342.10	5486.25	1374.71	25,862.90
2011	7348.74	2957.74	4527.22	1168.77	5081.93	1351.72	22,436.11

Notes: The income of tourism sectors in the table was obtained by adding the income of the international and domestic tourism sectors. The income of the international tourism sectors came from the China Statistical Yearbook. By obtaining the exchange rate of CNY to USD in the study year, the USD of the international tourism income in the study year was converted into CNY. The income of the domestic tourism sectors was calculated by the consumption proportion of each sector given in China’s Domestic Tourism Sample Survey Data and was multiplied by the total income.

**Table 3 ijerph-18-01218-t003:** Calculated value of *u* in 2011–2017.

Year	2011	2012	2013	2014	2015	2016	2017
*T*	1	2	3	4	5	6	7
*U*	0.0488	0.0952	0.1393	0.1813	0.2212	0.2592	0.2953

Note: a is 0.05.

**Table 4 ijerph-18-01218-t004:** Model statistics and parameter evaluation.

Equation	Model Summary					Parameter			
	R^2^	F	df1	df2	Sig.	Constant	b1	b2	b3
Linear	0.9668	145.67	1	5	0.00	13,449.59	129,056.15		
Logarithmic	0.8265	23.82	1	5	0.00	67,630.82	16,695.71		
Reciprocal	0.6172	8.06	1	5	0.04	48,300.51	−1512.71		
Quadratic	0.9981	1055.11	2	4	0.00	20,956.71	16,428.10	326,334.50	
Cubic	0.9982	548.35	3	3	0.00	21,678.82	−1861.93	448,942.86	237,473.50
Power	0.9011	45.55	1	5	0.00	86,017.36	0.48		
S	0.7147	12.53	1	5	0.02	10.81	−0.05		
Exponential	0.9939	818.47	1	5	0.00	18,262.32	3.63		

**Table 5 ijerph-18-01218-t005:** Potentials and contribution percentage of GHG emission reductions in different sub-scenarios and sectors.

**Emission Reduction** **(Mt CO_2e_)**		**2020**	**2030**	**2040**	**2050**
		65.71	420.04	534.38	785.79
Sub-scenarios	CES	54.55%	55.70%	52.15%	45.45%
	LCB	20.13%	24.31%	25.57%	28.96%
	ESF	17.82%	13.77%	13.70%	13.52%
	DSO	14.06%	17.79%	23.06%	32.24%
Sectors	Transportation	80.04%	88.41%	89.83%	90.48%
	Accommodation	4.49%	2.35%	1.92%	1.75%
	Catering	8.38%	4.64%	4.06%	3.72%
	Sightseeing	0.68%	0.52%	0.51%	0.51%
	Shopping	4.84%	2.51%	2.10%	1.90%
	Other	1.08%	0.95%	0.93%	0.90%
	Waste	0.49%	0.61%	0.65%	0.75%

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
