# Peer review of "LEAP-Based Greenhouse Gases Emissions Peak and Low Carbon Pathways in China’s Tourist Industry"

_ijerph, 2021, doi:10.3390/ijerph18031218_

Round 1

Reviewer 1 Report

Dear Dandan Liu, Dewei Yang and Anmin Huang!

I am sincerely grateful to you for the original thematic combination of environmental issues and tourism in my articles.

In my opinion, the article meets all the requirements for scientific articles.

The article defines the goal, theoretical and empirical literature is reviewed, appropriate econometric tools were used.

The article takes into account the conclusions of the research and refers to the polemics and opinions on the topic under study.

The paper wis well written and quite professional.

I support publishing a peer-reviewed article.

The only minor improvements I suggest: two small remarks

- The article does not provide a forecast for the impact of the pandemic on solving this problem. You are only one sentence focused on this issue. It would be interesting to get acquainted with certain calculations in this regard. But this is purely advisory in nature and does not affect the quality of research conducted by you. Perhaps this will be described in your next article.

- it would be good to indicate the source of borrowing statistical material, for example under Table 2., it might be useful for some of the readers.

Thank You for the opportunity to review this interesting paper.

Good luck in your further scientific work.

With gratitude,

dr Viktor Trynchuk

Author Response

Response to Reviewer 1 Comments We are pleased to note the favorable comments of reviewers in their opening sentence. In new MS, we have further explained and revised the questions you concerned. Thank you for your constructive comments. Point 1: The article does not provide a forecast for the impact of the pandemic on solving this problem. You are only one sentence focused on this issue. It would be interesting to get acquainted with certain calculations in this regard. But this is purely advisory in nature and does not affect the quality of research conducted by you. Perhaps this will be described in your next article. Response 1: In the new manuscript, we have added some new explanations to illustrate the impact of the pandemic, however, a more detailed study is expected in the next step. Thank you for your suggestion. At the beginning of 2020, COVID-19 had a great impact on the global tourist industry [34]. The lock-down measures by China to cope with COVID-19 pushed many small and medium-sized tourism industries on the verge of bankruptcy. Therefore, according to the actual situation, the GHG emissions of the tourist industry in 2020 will be lower than the predicted value in this study. But at this time, tourism demand is only temporarily suppressed, not disappeared. With the control of the epidemic situation, the tourist industry in China is recovering gradually and the National Day holiday in October received 637 million domestic visitors [35]. The GHG emissions of the tourist industry will still show an increasing trend in the future. Of course, the new demand and momentum brought by the epidemic also affect the tourism industry. The innovation of science and technology represented by internet, big data and artificial intelligence forces the reform of tourism industry. The "cloud tourism" promoted by live broadcasting has greatly enriched people's daily life and leisure activities. The series of reforms brought about by this epidemic are undoubtedly green. Point 2: It would be good to indicate the source of borrowing statistical material, for example under Table 2., it might be useful for some of the readers. Response 2: We followed your suggestion and added a note at the bottom of Table 2. Notes:The income of tourism sectors in the table is obtained by adding the income of international and domestic tourism sectors. The income of international tourism sectors comes from China Statistical Yearbook. By obtaining the exchange rate of CNY to USD in the study year, the USD of international tourism income in the study year is converted into CNY. The income of domestic tourism sectors is calculated by the consumption proportion of each sector given in China's Domestic Tourism Sample Survey Data is multiplied by the total income.

Reviewer 2 Report

It is an interesting paper on a very actual topic. The methodology used is not exceptionally sophisticated but correct.

The only improvement that I think should be raised is in conclusions. The conclusion section is not long and it would be good to complete it indicating possible future analyzes, possible improvements or adaptations of the proposed and applied model. You could even carry out a little self-criticism and propose some limitation of the proposed model.

Figures would need to adjust the size.

Author Response

Response to Reviewer 2 Comments

We should like to thank the referees for their helpful comments and hope that we have now produced a more balance and better account of our work.

Point 1: The only improvement that I think should be raised is in conclusions. The conclusion section is not long and it would be good to complete it indicating possible future analyzes, possible improvements or adaptations of the proposed and applied model. You could even carry out a little self-criticism and propose some limitation of the proposed model.

Response 1: We followed your suggestions and added some limitations and possible improvements of future study in conclusions.

In short, the concept of low-carbon tourism development has now been accepted. GHG prediction can better set the peak target of tourism and the pace of emission-reduction, and provide more effective suggestions for the sustainable development of tourism. Just as any case study, there are some limitations and possible improvements of future study: (1) Data is still the main restricting factor of tourism carbon emission re-search. Due to the difficulty of data acquisition, this study uses tourism income as the main driving factor. In the next step of research, it can obtain the specific technical parameters of different types of tourism, catering, accommodation, transportation, such as the number of beds, electrical type, power consumption and so on. (2) Carbon sink will be included in the measurement of tourism carbon emissions, such as water wetland, wood-land, arable land, pastoral, grassland and other natural resources, as well as the carbon compensation of tourism stakeholders. (3) It is necessary to consider the cost and technical difficulty of emission reduction measures, and to improve the carbon neutral path of the tourist industry from economic, social and environmental aspects.

Point 2: Figures would need to adjust the size.

Response 2: In the new manuscript, we have enlarged the text in the figure to improve readability. You can see the Figure 2,3,4,5,6.

Reviewer 3 Report

The work seems to me of great interest, and I think it should be published. The reasons that seem relevant to me are the following: 1. The application of a methodology that provides convincing results. 2. The focus on the environmental issue, that is, on the externalities generated by an economic activity that apparently seemed to have few ecological impacts. 3. The use of indicators other than the strictly chrematistic ones.

Now, to be concise, I detect some deficiencies that have not been sufficiently clarified in the text: a) The formula that is applied is very confusing, and its explanation as well. The importance of this equation is not understood, especially when the monetary units are provided in national currency (it would be important, in my opinion, if there was an explanation of exchange rates in this sense). b) It seems extremely risky for me to make such long-term predictions (until 2040!) taking into account the international demand scenario in which we operate. This is equally ascribable to Chinese national tourism, if this is the authors' approach. I would be much more cautious with these long-term forecasts and in a scenario of great uncertainty. c) It would have been interesting to have a table of biophysical indicators, since the authors take great pains to work on variables that are not strictly chrematistic. For example: energy consumption in TEP, CO2 emissions per capita, production of urban solid waste, consumption of fresh water, to name just a few vectors that other authors have worked on.

Author Response

Response to Reviewer 3 Comments

In new MS, we reorganize and rewrite most part of paper to make our paper logically readable. We hope the new MS has addressed the scientific basis with new reference criteria and all what you concern. Thank you for your constructive comments.

Point 1: The formula that is applied is very confusing, and its explanation as well. The importance of this equation is not understood, especially when the monetary units are provided in national currency (it would be important, in my opinion, if there was an explanation of exchange rates in this sense).

Response 1: In the new manuscript, we update the explanation of the formula. We should like to thank the referees for their helpful comments and hope that we have now produced a more balance and better account of our work. You can see the chapter 3.2 dan 3.3. In addition, different numbers will lead to different formulas. The coefficient of formula 2 cannot simply be labeled with exchange rates, and they may get very different results.

Point 2: It seems extremely risky for me to make such long-term predictions (until 2040!) taking into account the international demand scenario in which we operate. This is equally ascribable to Chinese national tourism, if this is the authors' approach. I would be much more cautious with these long-term forecasts and in a scenario of great uncertainty.

Response 2: Actually, the future is indeed unpredictable with great uncertainty. Including the COVID-19, many countries and regions have stagnated their economies. Prediction is also difficult to perfectly reflect the reality, but prediction is a warning and a reference, which can help decision makers choose more reasonable solutions. This paper predicts that the time to 2040 is due to China's commitment to reach the peak of carbon emissions by 2030. This paper sets a longer time because the characteristics of tourist industry are likely to have no peak or peak delay. We hope to determine whether there is a peak in the tourist industry and when the peak time will be, so as to really adjust the emission reduction strategy.

Point 3: It would have been interesting to have a table of biophysical indicators, since the authors take great pains to work on variables that are not strictly chrematistic. For example: energy consumption in TEP, CO2 emissions per capita, production of urban solid waste, consumption of fresh water, to name just a few vectors that other authors have worked on.

Response 3: Thank you for your suggestions, the focus of this paper is to study the impact of policy measures on tourism energy conservation and emission reduction, so many vectors are incomplete. Secondly, it is difficult to collect detailed data on a macro scale due to the limitations of data. The next step may be to study the economic costs of emission reduction measures and other related factors, which will also be added to the limitations and possible improvements of future study in conclusions.